# Efficient and Light-Weight Federated Learning via Asynchronous Distributed Dropout

**Chen Dun**[1]  **Mirian Hipolito**[2]  **Chris Jermaine**[1]  **Dimitrios Dimitriadis**[2]  **Anastasios Kyrillidis**[1]

## Abstract

Asynchronous learning protocols have regained attention lately, especially in the Federated Learning (FL) setup, where slower clients can severely impede the learning process. Herein, we propose `AsyncDrop`, a novel asynchronous FL framework that utilizes dropout regularization to handle device heterogeneity in distributed settings. Overall, `AsyncDrop` achieves better performance compared to state of the art asynchronous methodologies, while resulting in less communication and training time overheads. We implement our approach and compare it against other asynchronous baselines, both by design and by adapting existing synchronous FL algorithms to asynchronous scenarios. Empirically, `AsyncDrop` reduces the communication cost and training time, while matching or improving the final test accuracy in diverse non-i.i.d. FL scenarios.

## 1 Introduction

**Background on Federated Learning.** Federated Learning (FL) (McMahan et al., 2017; Li et al., 2018; Karimireddy et al., 2019) is a distributed learning protocol that has witnessed fast development the past demi-decade. FL deviates from the traditional distributed learning paradigms and allows the integration of edge devices —such as smartphones (Stojkovic et al., 2022), drones (Qu et al., 2021), and IoT devices (Nguyen et al., 2021)— in the learning procedure.

Yet, such real-life, edge devices are extremely *heterogeneous* (Wang et al., 2021): they have drastically different specifications in terms of compute power, device memory and achievable communication bandwidths. Directly applying common *synchronized* FL algorithms –such as FedAvg and FedProx (Li et al., 2018; McMahan et al., 2017) that require full model broadcasting and global synchronization– results often in a "stragglers" effect (Nguyen et al., 2022; Huba et al., 2022; Tandon et al., 2017).

**The ubiquitous synchronous training.** One way to handle such issues is by utilizing *asynchrony* instead of *synchrony* in the learning process. To explain the main differences, let us first set up the background. In a synchronous distributed algorithm, a global model is usually stored at a central server and is broadcast periodically to all the participating devices. Then, each device performs local training steps on its own model copy, before the device sends the updated model to the central server. Finally, the central server updates the global model by aggregating the received model copies. This protocol is followed in most FL algorithms, including the well-established FedAvg (McMahan et al., 2017), FedProx (Li et al., 2018), FedNova (Wang et al., 2020) and

SCAFFOLD (Karimireddy et al., 2019). The main criticism against synchronous learning could be that it often results in heavy communication/computation overheads and long idle/waiting times for workers.

**Asynchrony and its challenges.** The deployment of a asynchronous learning method is often convoluted. In the past decade, HogWild! (Niu et al., 2011; Liu et al., 2014) has emerged as a general asynchronous distributed methodology, and has been applied initially in basic ML problems like sparse linear/logistic regression (Zhuang et al., 2013; Yun et al., 2013; Hsieh et al., 2015). Ideally, based on sparsity arguments, each edge device can independently update parts of the global model –that overlap only slightly with the updates of other workers– in a lock-free fashion (Niu et al., 2011; Liu et al., 2014). This way, faster, more powerful edge workers do not suffer from idle waiting due to slower stragglers. Yet, the use of asynchrony has been a topic of dispute in distributed neural network training (Dean et al., 2012; Chen et al., 2016).

**Resurgence in asynchrony.** Recently, asynchronous methods have regained popularity, mainly due to the interest in applying asynchronous motions within FL on edge devices. Yet, traditional off-the-shelf asynchronous algorithms still have issues, which might be exacerbated in the FL setting. As slower devices take longer local training time, this might result in inconsistent update schedules of the global model, compared to that of faster devices. This might have ramifications: $i$) For FL on i.i.d. data, this will cause the gradient staleness problem and result in convergence rate decrease; and, $ii$) on non-i.i.d. data, this will result in a significant drop in global model final accuracy.

As solutions, novel approaches on asynchronous FL propose weighted global aggregation techniques that take into consideration the heterogeneity of the devices (Xie et al., 2019; Chen et al., 2019; Shi et al., 2020); yet, these methods often place a heavy computation/communication burden, as they rely on broadcasting full model updates to all the clients and/or the server. Other works monitor client speed to guide the training assignments (Li et al., 2021; Chai et al., 2020). Finally, recent efforts propose semi-asynchronous methods, where participating devices are selected and buffered in order to complete a semi-synchronous global update periodically (Huba et al., 2022; Wu et al., 2020).

**What is different in this work?** As most algorithms stem from adapting asynchrony in synchronous FL, *one still needs to broadcast the full model to all devices, following a data parallel distributed protocol (Farber & Asanovic, 1997; Raina et al., 2009), regardless of device heterogeneity.* This inspire us to ask a key question:

> *"Can we select submodels out of the global model and send these instead to each device, taking into account the device heterogeneity?"*

We answer this question affirmatively, by proposing a novel distributed dropout method for FL. We dub our method `AsyncDrop`. Our approach assigns different submodels to each device[1]; empirically, such a strategy decreases the required time to converge to an accuracy level, while preserving favorable final accuracy. Our idea is based on the ideas of HogWild! (Niu et al., 2011; Liu et al., 2014) –in terms of sparse submodels– and Independent Subnetwork Training (IST) (Yuan et al., 2022; Dun et al., 2022; Liao & Kyrillidis, 2021; Wolfe et al., 2021) –where submodels are deliberately created for distribution, in order to decrease both computational and communication requirements.

Overall, the contributions of this work can be summarized as follows:

- We consider and propose *asynchronous distributed dropout* (`AsyncDrop`) for efficient large-scale FL. Our focus is on non-trivial, non-convex ML models –as in neural network training– and our framework provides specific engineering solutions for these cases in practice.

- We theoretically characterize and support our proposal with rigorous and non-trivial convergence rate guarantees. Currently, our theory assumes bounded delays; our future goal is to exploit recent developments that drop such assumptions (Koloskova et al., 2022). Yet, our theory already considers the harder case of neural network training, which is often omitted in existing theory results.

- We provide specific implementation instances and share

---

[1]We consider both random assignment, as well as structured assignments, based on the computation power of the devices.

"best practices" for faster distributed FL in practice. As a side-product, our preliminary results include baseline asynchronous implementations of many synchronous methods (such as FedAvg, FedProx, and more), that are not existent currently, to the best of our knowledge.

## 2 PROBLEM SETUP AND CHALLENGES

**Optimization in neural network training.** We consider FL scenarios over *supervised* neural network training: i.e., we optimize a loss function $\ell(\cdot, \cdot)$ over a dataset, such that the model maps unseen data to their true labels, unless otherwise stated. For clarity, the loss $\ell(\mathbf{W}, \cdot)$ encodes both the loss metric and the neural architecture, with parameters $\mathbf{W}$. Formally, given a data distribution $\mathcal{D}$ and $\{\mathbf{x}_i, y_i\} \sim \mathcal{D}$, where $\mathbf{x}_i$ is a data sample, and $y_i$ is its corresponding label, classical deep learning aims in finding $\mathbf{W}^\star$ as in:

$$\mathbf{W}^\star = \underset{\mathbf{W} \in \mathcal{H}}{\arg\min} \left\{ \mathcal{L}(\mathbf{W}) := \tfrac{1}{n} \sum_{i=1}^{n} \ell(\mathbf{W}, \{\mathbf{x}_i, y_i\}) \right\},$$

where $\mathcal{H}$ denotes the model hypothesis class that "molds" the trainable parameters $\mathbf{W}$.

The minimization above can be achieved by using different approaches, but almost all training is accomplished via a variation of *stochastic gradient descent* (SGD) (Robbins & Monro, 1951). SGD modifies the current guess $\mathbf{W}_t$ using stochastic directions $\nabla \ell_{i_t}(\mathbf{W}_i) := \nabla \ell(\mathbf{W}_i, \{\mathbf{x}_{i_t}, y_{i_t}\})$. I.e., $\mathbf{W}_{t+1} \leftarrow \mathbf{W}_t - \eta \nabla \ell_{i_t}(\mathbf{W}_t)$. Here, $\eta > 0$ is the learning rate, and $i_t$ is a single or a mini-batch of examples. Most FL algorithms are based on these basic stochastic motions, like FedAvg (McMahan et al., 2017), FedProx (Li et al., 2018), FedNova (Wang et al., 2020) and SCAFFOLD (Karimireddy et al., 2019).

**FL formulation.** Let $S$ be the total number of clients in a distributed FL scenario. Each client $i$ has its own local data $\mathcal{D}_i$ such that the whole dataset satisfies $\mathcal{D} = \cup_i \mathcal{D}_i$, and usually $\mathcal{D}_i \cap \mathcal{D}_j = \emptyset, \forall i \neq j$. The goal of FL is to find a global model $\mathbf{W}$ that achieves good accuracy on all data $\mathcal{D}$, by minimizing the following optimization problem:

$$\mathbf{W}^\star = \underset{\mathbf{W} \in \mathcal{H}}{\arg\min} \left\{ \mathcal{L}(\mathbf{W}) := \tfrac{1}{S} \sum_{i=1}^{S} \ell(\mathbf{W}, \mathcal{D}_i) \right\},$$

where $\ell(\mathbf{W}, \mathcal{D}_i) = \frac{1}{|\mathcal{D}_i|} \sum_{\{\mathbf{x}_j, y_j\} \in \mathcal{D}_i} \ell(\mathbf{W}, \{\mathbf{x}_j, y_j\})$. Herein, we consider both i.i.d. and non-i.i.d. cases, since local data distribution $\mathcal{D}_i$ can be heterogeneous and follow a non-i.i.d. distribution.

**Details of asynchronous training.** An abstract description of how asynchronous FL operates is provided in Algorithm 1. In particular, given a number of server iterations $T$, each client $i$ gets the updated global model $\mathbf{W}_t$ from the server, and further locally trains it using $\mathcal{D}_i$ for a number of local

**Algorithm 1** Meta Asynchronous FL

**Parameters**: $T$ iters, $S$ clients, $l$ local iters., $\mathbf{W}$ as current global model, $\mathbf{W}_i$ as local model for $i$-th client, $\alpha \in (0, 1)$, $\eta$ step size.

———————— $\infty$ ————————

$\mathbf{W} \leftarrow$ randomly initialized global model.
//On each client $i$ asynchronously:
**for** $t = 0, \ldots, T - 1$ **do**
  $\mathbf{W}_{i,t} \leftarrow \mathbf{W}$
  //Train $\mathbf{W}_i$ for $l$ iters. via SGD
  **for** $j = 1, \ldots, l$ **do**
    $\mathbf{W}_{i,t} \leftarrow \mathbf{W}_{i,t} - \eta \frac{\partial \mathcal{L}}{\mathbf{W}_{i,t}}$
  **end for**
  //Write local to global model
  $\mathbf{W} \leftarrow (1 - \alpha) \cdot \mathbf{W} + \alpha \cdot \mathbf{W}_{i,t}$
**end for**

iterations $l$. Asynchronous FL assumes each client has different computation power and communication bandwidth; this can be abstracted by different wall-clock times required to finish a number of local training iterations. Thus, when client $i$ has completed its round, the updated model is shared with the server to be aggregated, before the next round of communication and computation starts for client $i$. *This is different from classical synchronous FL, where the global model is updated only when all participating clients finish (or time-out) certain local training iterations.*

## 3 RELATED WORK

**Challenges.** Asynchronous steps often lead to inconsistent update schedules of the global model and are characterized by gradient staleness and drifting.

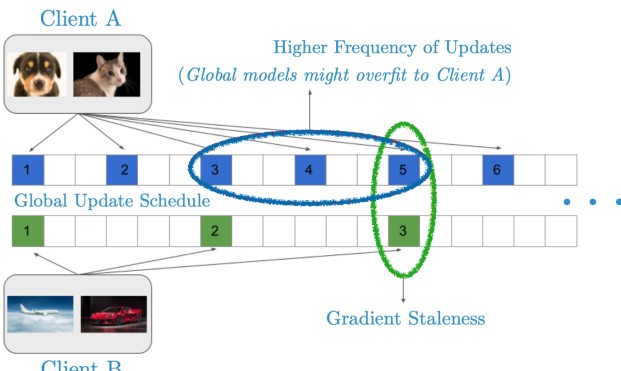

*Figure 1.* Potential issues in asynchronous FL.

Consider the toy setting in Figure 1. The two clients (Clients A and B) have a significantly different update schedule on the global model: Here, Client A has higher computational power or communication bandwidth –compared to client B– potentially leading to model drifting, lack of fair training and more severe gradient staleness. On top, consider these two

clients having different local (non-i.i.d.) data distributions.

**Related Work.** The issue of model drifting due to data "non-iidness" is a central piece in FL research. Algorithms, such as FedProx (Li et al., 2018), utilize regularization to constrain local parameters "drifting" away.

The gradient staleness problem has been widely studied in asynchronous FL (Xie et al., 2019; Chen et al., 2019; Shi et al., 2020; Li et al., 2021; Chai et al., 2020). In these approaches, the weight of each local client update is proportional to the "capabilities" of the client. This should decrease the negative impact from stale gradients by slower clients. Semi-asynchronous methods have been proposed (Huba et al., 2022; Wu et al., 2020); yet, they require fast clients to wait until all other clients' updates are completed, in order to receive the updated model for the next round.

Finally, numerous quantization (Alistarh et al., 2017; Yu et al., 2019) and sparsification (Aji & Heafield, 2017; Jiang & Agrawal, 2018) techniques have been proposed for reducing computation and communication costs in FL.

## 4 ASYNCH. DISTRIBUTED DROPOUT

**(Distributed) Dropout**. Dropout (Wan et al., 2013; Srivastava et al., 2014; Gal & Ghahramani, 2016; Courbariaux et al., 2015) is a widely-accepted regularization technique. The procedure of Dropout is as follows: per training round, a random mask over the parameters is generated; this mask is used to nullify part of the neurons in the neural network for this particular iteration. Variants of dropout include the drop-connect (Wan et al., 2013), multisample dropout (Inoue, 2019), Gaussian dropout (Wang & Manning, 2013), and the variational dropout (Kingma et al., 2015).

The idea of dropout has also been used in efficient distributed and/or FL scenarios. (Horvath et al., 2021a) introduces FjORD and the Ordered Dropout, a *synchronous* distributed dropout technique that leads to ordered, nested representation of knowledge in models, and enables the extraction of lower footprint submodels without the need of retraining. Such submodels are more suitable in client heterogeneity, as they adapt submodel's width to the client's capabilities. See also Nested Dropout (Rippel et al., 2014) and HeteroFL (Diao et al., 2020).

**Our proposal and main hypothesis**. We focus on the *asynchronous version of distributed dropout.* In Appendix, we study theoretically whether asynchrony provably works in non-trivial non-convex scenarios –as in training neural networks– with random masks that generate submodels for each worker. The algorithm is described in Algorithm 2, dubbed as AsyncDrop, and is based on recent distributed protocols (Yuan et al., 2022; Dun et al., 2022; Liao & Kyrillidis, 2021; Wolfe et al., 2021); key features are highlighted

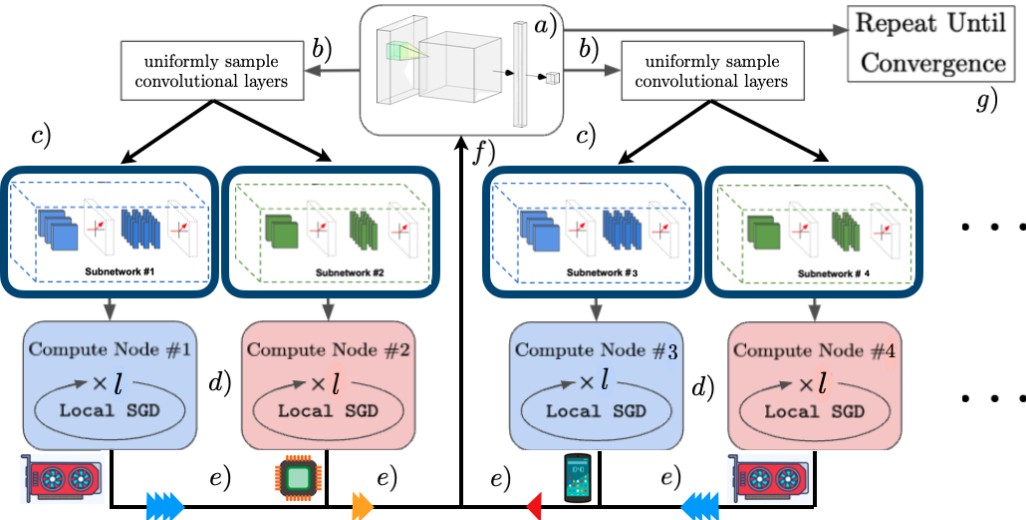

*Figure 2.* Schematic representation of `AsyncDropout`. *a)* This is a simple representation of a CNN model. Our algorithm applies for arbitrary depth of CNNs (ResNets) as well as other architectures (MLPs, LSTMs, etc); here we restrict to a shallow CNN for illustration purposes. *b)* Per request, random sub-sampled CNN models are created that result into different *subnetworks*. *c)* These submodels are distributed to devices with different computational capabilities (here GPU, CPU, or a smartphone). *d)* Without loss of generality, we assume that all devices train locally the submodel for *l* iterations. *e)* However, each device finishes local training in different timestamps (shown as different colored arrows: red: slow speed; orange: moderate speed; blue: fast speed). *f)* Yet, the global model is asynchronously updated and new submodels are created without global synchronization. *g)* The above procedure is repeated till convergence.

---

**Algorithm 2** Asynchronous dropout (`AsyncDrop`)

---

**Parameters**: $T$ iters, $S$ clients, $l$ local iters., $\mathbf{W}$ as current global model, $\mathbf{W}_i$ as local model for $i$-th client, $\alpha \in (0, 1)$, $\eta$ step size.

———————— ∞ ————————

$\mathbf{W} \leftarrow$ randomly initialized global model.
//On each client $i$ asynchronously:
**for** $t = 0, \ldots, T - 1$ **do**
  Generate mask $\mathbf{M}_{i,t}$
  $\mathbf{W}_{i,t} \leftarrow \mathbf{W}_t \odot \mathbf{M}_{i,t}$
  //Train $\mathbf{W}_{i,t}$ for $l$ iters. via SGD
  **for** $j = 1, \ldots, l$ **do**
    $\mathbf{W}_{i,t} \leftarrow \mathbf{W}_{i,t} - \eta \frac{\partial \mathcal{L}}{\mathbf{W}_{i,t}}$
  **end for**
  //Write local to global model
  $\mathbf{W}_{t+1} \leftarrow \mathbf{W}_t \odot (\mathbf{M}_{i,t})^c$
  $\qquad\qquad + ((1 - \alpha) \cdot \mathbf{W}_t + \alpha \cdot \mathbf{W}_{i,t}) \odot \mathbf{M}_{i,t}$
**end for**

---

in teal-colored text. The main difference from Algorithm 1 is that Algorithm 2 splits the model vertically per iteration, where each submodel contains all layers of the neural network, but only with a (non-overlapping) subset of neurons being active in each layer. Multiple local SGD steps can be performed without the need for the workers to communicate. See also Figure 2 for a schematic representation of asynchronous distributed dropout for training a CNN.

## 5 EXPERIMENTS

**Setup.** We generate simulated FL scenarios with 104 clients/devices of diverse computation and communication capabilities. We implement clients as independent processes, each distributed on different GPUs with access to the same RAM space. We follow HogWild!'s distributed model (Niu et al., 2011): *i)* we use a shared-memory system to store the global model; *ii)* each simulated client can update/read the global model in a fully lock free mode; and *iii)* each client transfers the local model to the assigned GPU for local training. We activate 8 clients at any given moment.

We use 25% dropout rate in (`Hetero`) `AsyncDrop` for CNN and MLP, while we use 12.5% for LSTM. *Even for such low dropout rates, the gains in training are obvious and significant, as we show in the experiments.* In the appendix, we provide ablation studies on how the dropout rate affects the performance of `AsyncDrop`-family of algorithms. We simulate the communication and computation savings by inserting shorter time delay, based on which we estimate the training time and communication cost.

**Simulation of heterogeneous computations.** We abstract heterogeneous computation and communication capabilities by "forcing" different delays after each training iteration. The delay time is inverse proportional to the intended capacity. In our experiments, we simulated 8 levels of computation and communication capabilities, that are evenly distributed between the slowest client and fastest clients.

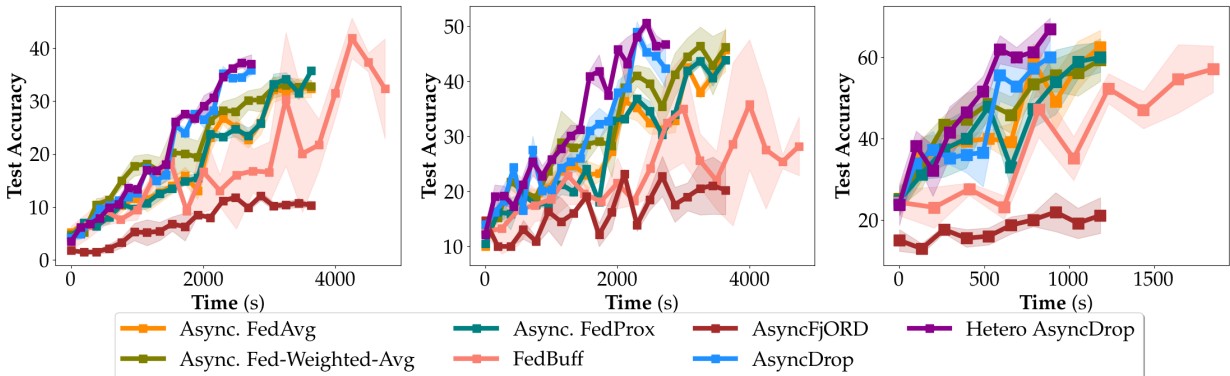

*Figure 3.* ResNet-based model. ***Left:*** CIFAR100 non-i.i.d.; ***Middle:*** CIFAR10, non-i.i.d.; ***Right:*** FMNIST, non-i.i.d.

The difference between the slowest and the fastest clients is selected to be $\sim 5\times$. We make sure that, at any given moment, clients with diverse capacity are active. Finally, all clients with similar computation power shall have similarly biased local data distribution.[2]

**Problem cases.** We experiment on diverse neural network architectures and diverse types of learning tasks, including ResNets on Computer Vision datasets (CIFAR10, CIFAR100, FMNIST), MLPs on FMNIST dataset, and LSTMs on sentimental analysis (IMDB).

**Baseline methods.** For comparison, the baselines we consider are: $i$) the asynchronous FedAvg is the direct adaptation of FedAvg with asynchronous motions; $ii$) the asynchronous FedAvg with weighted aggregation represents the general approach of assigning devices different "importance", based on their capabilities (Xie et al., 2019; Chen et al., 2019; Shi et al., 2020); $iii$) the asynchronous FjORD is our asynchronous adaptation of (Horvath et al., 2021b); $iv$) the asynchronous FedProx adds an independent proximal loss to the local training loss of each device, in order to control gradient staleness and overfiting (Li et al., 2018); and $v$) FedBuff is a semi-asynchronous method which uses buffers for stale updates in a synchronized global scheme (Nguyen et al., 2022).

For all baselines and (Hetero) AsyncDrop on CIFAR10, CIFAR100, FMNIST, we set the local iterations at $l = 50$ while for IMDB, $l = 40$. For FedBuff, we set the buffer size to 4, which is half of the activated clients. We perform 3 trials with different random seeds. We report the maximum test accuracy, as well as the estimated time and communication cost to reach a certain target accuracy: *we select the second lowest test accuracy among all baselines as the target accuracy (third column in result tables).*

**CNN-based results.** We test (Hetero) AsyncDrop on ResNet34 and using CIFAR10, CIFAR100 and FMNIST

datasets. For the CIFAR10 and CIFAR100 datasets, we train for 320 epochs, while for the FMNIST dataset, we train for 160 epochs. We stop the execution when the fastest client finishes all its epochs. As shown in Table **??**, Hetero AsyncDrop shows non-trivial improvements in *final accuracy, training time and communication cost, simultaneously.* Hetero AsyncDrop shows lower accuracy compared with FedBuff in the CIFAR100 case; yet, it achieves up to $85\%$ reduction in training time, due to the fact that FedBuff will require faster workers to wait until the buffer is filled to update the global model (this also justifies the up to $\sim 42.22\%$ reduction in total communication cost). Finally, we observe that (Hetero) AsyncDrop shows quite stable performance; in red color we indicate the variability of results over trials. The similar training time of some baselines to reach target accuracy is caused by epoch-wise testing, using same epoch-wise learning rate schedule and similar convergence rate as shown in Figure 3.

**MLP-based and LSTM-based results.** We adapt the (Hetero) AsyncDrop to the MLP model by applying the hidden neuron Dropout, which is similar to channel dropout in CNNs (160 epochs). For the LSTM and the IMDB sentimental analysis dataset (80 epochs), we create non-i.i.d. datasets based on different label distribution in each local training set. As shown in Table 1 in the Appendix, Hetero AsyncDrop achieves better performance overall over the MLP model, in terms of accuracy, training time and communication cost. As shown in Table 2 in the Appendix, for the LSTM model, Hetero AsyncDrop shows comparable accuracy with respect to other baselines, while achieving reduction in training time and communication cost in most cases. Our conjecture for the lower gain compared with other architectures is that LSTM-based (or even RNN-based) architectures might be difficult to our proposed dropout score mechanism, as the update of each network parameter is the average of several virtual parameters in the unrolled network.

---

[2]This setting is to avoid the fastest clients with similar computation power cover all the data, which will reduce the problem into a trivial synchronous federated learning problem.

---

[*]Equal contribution  [1]Department of Computer Science Rice University  [2]Microsoft Research Lab. **AUTHORERR: Missing \mlsyscorrespondingauthor.**

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

**Theoretical Results**. We are interested in understanding whether such a combination of asynchronous computing and dropout techniques lead to convergence and favorable results: *given the variance introduced by both asynchronous updates and training of submodels, it is not obvious whether –and under which conditions– such a protocol would work.*

For ease of presentation and clarity of results, we analyse a one-hidden-layer CNN, and show convergence with random filter dropout. Consider a training dataset $(\mathbf{X}, \mathbf{y}) = \{(\mathbf{x}_i, y_i)\}_{i=1}^n$, where each $\mathbf{x}_i \in \mathbb{R}^{\hat{d} \times p}$ is an image and $y_i$ being its label. Here, $\hat{d}$ is the number of input channels and $p$ the number of pixels. Let $q$ denote the size of the filter, and let $m$ be the number of filters in the first layer. Based on previous work (Du et al., 2019), we let $\hat{\phi}(\cdot)$ denote the patching operator with $\hat{\phi}(x) \in \mathbb{R}^{q\hat{d} \times p}$. Consider the first layer weight $\mathbf{W} \in \mathbb{R}^{m \times q\hat{d}}$, and second layer (aggregation) weight $\mathbf{a} \in \mathbb{R}^{m \times p}$. We assume that only the first layer weights $\mathbf{W}$ is trainable. The CNN trained on the means squared error has the form:

$$f(\mathbf{x}, \mathbf{W}) = \left\langle \mathbf{a}, \sigma\left(\mathbf{W}\hat{\phi}(\mathbf{x})\right), ; \right\rangle, \mathcal{L}(\mathbf{W}) = \|f(\mathbf{X}, \mathbf{W}) - \mathbf{y}\|_2^2,$$

where $f(\mathbf{x}, \cdot)$ denotes the output of the one-layer CNN for input $\mathbf{x}$, and $\mathcal{L}(\cdot)$ is the loss function. We use the $\ell_2$-norm loss for simplicity. We make the following assumption on the training data and the CNN weight initialization.

**Assumption .1** *(Training Data) Assume that for all $i \in [n]$, we have $\|\mathbf{x}_i\|_F = q^{-\frac{1}{2}}$ and $|y_i| \leq C$ for some constant $C$. Moreover, for all $i, i' \in [n]$ we have $\mathbf{x}_i \neq \mathbf{x}_{i'}$.*

Note that this can be satisfied by normalizing the data. For simplicity of the analysis, let $d := q\hat{d}$.

**Assumption .2** *(Initialization)* $\mathbf{w}_{0,i} \sim \mathcal{N}\left(0, \kappa^2 \mathbf{I}\right)$ *and* $a_{i,i'} \sim \left\{\pm \frac{1}{p\sqrt{m}}\right\}$ *for $i \in [m]$ and $i' \in [p]$.*

In an asynchronous scenario, the neural network weight is updated with stale gradients due to the asynchronous updates, where $\delta_t$ is the delay at training step $t$. *We assume $\delta_t$ is bounded by a constant $E$.* Then, a simple version of gradient descent under these assumptions looks like:

$$\mathbf{W}_t = \mathbf{W}_t - \eta \nabla_{\mathbf{W}} \mathcal{L}\left(\mathbf{W}_{t-\delta_t}\right), \quad \delta_t \leq E,$$

where $\mathbf{W}_{t-\delta_t}$ indicates that the gradient is evaluated on a earlier version of the model parameters. Given the above, we provide the following guarantees:

**Theorem .1** *Let $f(\cdot, \cdot)$ be a one-hidden-layer CNN with the second layer weight fixed. Let $\mathbf{u}_t$ abstractly represent the output of the model after $t$ iterations, over the random selection of the masks. Let $E$ denotes the maximum gradient delay/staleness. Let $\xi$ denote the dropout rate ($\xi = 1$ dictates that all neurons are selected), and denote $\theta = 1 - (1-\xi)^S$ the probability that a neuron is active in at least one sub-network. Assume the number of hidden neurons satisfies $m = \Omega\left(\max\{\frac{n^4 K^2}{\lambda_0^4 \delta^2} \max\{n, d\}, \frac{n}{\lambda_0}\}\right)$ and the step size satisfies $\eta = O\left(\frac{\lambda_0}{n^2}\right)$. Let $\kappa$ be a proper initialization scaling factor, and it is considered constant. We use $\lambda_0$ to denote the smallest eigenvalue of the Neural Tangent Kernel matrix. Let Assumptions 1 and 2 be satisfied. Then, the following convergence rate guarantee is proved to be satisfied:*

$$\mathbb{E}_{\mathbf{M}_t}\left[\|\mathbf{u}_{t+1} - \mathbf{y}\|_2^2\right] \leq \left(1 - \frac{\theta\eta\lambda_0}{4}\right)^t \|\mathbf{u}_0 - \mathbf{y}\|_2^2$$

$$+ O\left(\frac{\theta\eta\lambda_0^3\xi^2\kappa^2 E^2}{n^2} + \frac{\xi^2(1-\xi)^2\theta\eta n^3\kappa^2 d}{m\lambda_0} + \frac{\eta^2\theta^2 n\kappa^2\lambda_0\xi^4 E^2}{m^4} + \frac{\xi^2(1-\xi)^2\theta^2\eta^2 n^2\kappa^2 d}{m^3\lambda_0}\right.$$

$$\left. + \frac{\xi^2(1-\xi)^2\theta^2\eta^2\kappa^2\lambda_0 E^2}{m^3} + \frac{\xi^2(1-\xi)^2\theta^2\eta^2 n^2\kappa^2 d}{m^2\lambda_0} + \frac{n\kappa^2\left(\theta - \xi^2\right)}{S}\right)$$

**Remark #1.** This theorem states that the sum of the expected weight differences in the $t$-th iteration (i.e., $\mathbb{E}_{\mathbf{M}_t}[\|\mathbf{u}_{t+1} - \mathbf{y}\|_2^2]$) converges linearly to zero, as dictated by the red term $-\left(1 - \frac{\theta\eta\lambda_0}{4}\right)^t \|\mathbf{u}_0 - \mathbf{y}\|_2^2$– up to an error neighborhood, denoted with the Big-Oh notation term on the right hand side of the expression. Focusing on the latter, there are two types of additive errors: $i)$ the orange-colored terms origin from the dropout analysis: the term $1 - \xi$ is often called as "dropout rate" (when $\xi = 0$, no neu-

rons are selected and the loss hardly decreases, while when $\xi = 1$, all neurons are selected, and the orange-colored terms disappear). $ii)$ the violet-colored terms origin from the asynchronous analysis: when $E = 0$ (i.e., we boil down to synchronous computations), these terms also disappear).

**Remark #2.** Beyond the above extreme cases, we observe that the error region terms can be controlled by algorithmic and model-design choices: e.g., when the size of the dataset $n$ increases, the first term $\frac{\theta\eta\lambda_0^3\xi^2\kappa^2 E^2}{n^2}$ can be controlled; for

sufficiently wide neural network, the terms with $m$ in the denominator can be made arbitrarily small; finally, notice that increasing the number of subnetworks $S$ will drive the last term in the bound zero.