# OpenReview forum: "Efficient and Light-Weight Federated Learning via Asynchronous Distributed Dropout"
_MLSys/2023/Workshop/RCLWN — MLSys-RCLWN 2023_

### Official Review · Reviewer_FKGd · 2023-04-25
**AsyncDrop has potential but requires further justification and refinement.**

**Rating:** 6
**Confidence:** 4

**Review:**

This paper proposes AsyncDrop, a method for addressing device (system) heterogeneity by incorporating dropout regularization in asynchronous Federated Learning (FL).


**Originality**: The paper is the first to suggest using dropout regularization in asynchronous FL to tackle system heterogeneity. Related work is appropriately cited.
Quality: The submission is technically sound and offers a theoretical guarantee. However, the experimental setup seems ad-hoc.

**Clarity**: Although the paper is well-written and organized, a more detailed description and rationale behind the main algorithm are expected.

**Significance**: While extending dropout regularization in asynchronous FL could prove valuable, the paper simply employs the naive aggregation optimizer and dropout mechanism, instead of directly applying asynchronous aggregation on dropout.

**Questions:**

1. Experimental settings: Several setups seem unconventional and lack proper citation, potentially appearing cherry-picked. a) Why are eight clients activated at any given time? This should resemble the training concurrency in Fedbuff, a crucial parameter for staleness. In Fedbuff[1] and [2], this value is typically set much higher (\~1000), leading to faster model updates. Further explanation is needed. b) Why are the number of local iterations unusually large (\~40), compared to related works (~10)? Providing references or reasoning would improve this. c) The number of epochs seems insufficient for model convergence. In Figure 3, models appear to require more epochs to converge. d) In Figure 3, Fedbuff's performance seems unstable, likely due to the hyperparameter setups. Aligning the setups with the original paper or providing a discussion would be beneficial.
2. Method: A more in-depth reasoning and description of the method are expected. a) How does mask random sampling adapt to different devices? b) In general, how does the algorithm accommodate varying devices?
3. Writing: Distinguishing between system and statistical heterogeneity in the contribution section will help prevent confusion.


[1] Federated learning with buffered asynchronous aggregation.

[2] Green Federated Learning

---

### Official Review · Reviewer_o7vg · 2023-04-27
**Review of Efficient and Light-Weight Federated Learning via Asynchronous Distributed Dropout**

**Rating:** 7
**Confidence:** 3

**Review:**

This work addresses the challenge of device heterogeneity in Federated Learning by proposing an asynchronous version of distributed dropout. It also provides theoretical guarantees of convergence rate for the proposed algorithm.

1. Where the results are very promising in terms of accuracy, communication, and time overhead, the key novelty from existing methods such s heteroFL is not clear to me. My understanding is the mask is changing every new round for this work whereas for HeteroFL it was fixed.

2. Further, I am curious to know how data is distributed across clients.

3. On line number 237, a reference to the table is missing.

4. Regarding Figure 3, my suggestion is to also have tables for CIFAR10 and CIFAR100 results in the format of Table 1 would also add clarity to understanding the gains.

---

### Official Review · Reviewer_pKYS · 2023-04-29
**EFFICIENT AND LIGHT-WEIGHT FEDERATED LEARNING VIA ASYNCHRONOUS DISTRIBUTED DROPOUT**

**Rating:** 7
**Confidence:** 4

**Review:**

This paper proposes an asynchronous FL framework named AsyncDrop to reduce communication and training time overheads. AsyncDrop utilizes dropout regularization to assign different submodels to each device and handle device heterogeneity in distributed settings. Compared with other asynchronous baselines, AsyncDrop achieved better communication efficiency and better accuracy.

Pros:
1. This paper studies an important efficiency problem in FL.
2. Paper presentation is good.
3. The idea of using dropout in asynchronous FL settings is interesting.

Cons and Questions:
1. Please provide more details and discussions about the parameter aggregation step, the last step in Algorithm 2 (lines 206 and 207). What does c mean for M_{i,t}^{c}?
2. About the method: if a random mask is generated over the entire model parameters, wouldn't it be possible that the part selected from the global model overlaps with the part selected from the i-th local model? In this case, what would happen if two local models select the same set of global model parameters for an update? Wouldn't it be possible that the faster client still has to wait for the slower one?
3. Despite the simplicity of dropout, AsynDrop doesn't seem to really solve the issues of asynchronous FL. The efficiency gains can also be partially attributed to fewer parameters update. It is recommended to compare it with a synchronous version of Dropout under FL.
4. The models and datasets being studied are too simple. It is recommended to test more advanced neural architectures such as Transformers and more practical and challenging tasks (in future works).

---

### Official Review · Reviewer_N52Z · 2023-04-29
**The authors propose Hetero AsyncDrop, a novel algorithm for asynchronous federated learning which reduces the communication overhead of a FL session by allowing participating clients to asynchronously update only a fraction of the global model parameters.**

**Rating:** 7
**Confidence:** 3

**Review:**

In standard federated learning (FL), multiple devices with heterogeneous datasets and computational capabilities train a global model without sharing their own private data. One issue in synchronous FL is that the time needed to compute a FL round is determined by the slowest node. Therefore, the authors present an algorithm based on asynchronous federated learning to reduce the impact of straggler nodes in a FL session. In particular, the paper introduces Hetero AsyncDrop, where clients employs the well known Dropout technique to update only a fraction of the model parameters and therefore reduce communication overhead and time to accuracy of the global model.

The use of Dropout in asynchronous FL is original and provides relevant insights on whether it is possible or not to improve the metrics of an asynchronous FL session through pruning. For synchronous FL, there are already some previous works on dropout to achieve this goal in FL (namely Federated Dropout and its variants), but that do not affect the novelty of the proposed algorithm.

In the following paragraphs, strengths and weaknesses of the proposed algorithm are listed.

It is not clear how the masks for the dropout are computed. In the introduction, it is stated that the authors "consider both random assignment, as well as structured assignments, based on the computation power of the devices". However, computational power-based dropout is not further discussed and seems to be in contrast with the experiment section where a constant dropout rate is used for all clients. Moreover, In Section 4 it is stated that "Algorithm 2 splits the model vertically per iteration, where each submodel contains all layers of the neural network, but only with a (non-overlapping) subset of neurons
being active in each layer", but the mask generation procedure to produce these kinds of non-overlapping masks is not presented.

Hetero AsyncDrop has been evaluated on five datasets (CIFAR100, CIFAR10, FMNIST, IMDB), three models (ResNet34, MLP and LSTM-based neural network) and two tasks (image classification and sentiment analysis). However, except for ResNet, most hyper-parameters are not listed, thus affecting reproducibility of the results. For instance, a description of the model, the employed learning rates, and how datasets have been split into their non i.i.d versions should be clearly stated.

The algorithm has been evaluated against multiple baselines, thus making the experiments section sound. However, most of the baselines are adaptations of synchronous algorithm to the asynchronous scenario, which might undermine the capabilities of such algorithms which were not designed for that case. The authors also compare the performance to FedBuff, which is an asynchronous aggregation method which trades off performance in order to achieve aggregation security. Therefore, the performance of FedBuff are considerably lower than AsyncDrop, which do not consider secure aggregation. The benefits of AsyncDrop could be less prominent if secure aggregation was taken into account.

The performance analysis shows improvements compared to the baseline in Figure 3. In particular, AsyncDrop provides better final accuracy and time to convergence in the image classification task. This is due to the fact that the updates from the client are smaller in size, since dropout is applied. The table with the performance analysis in the CNN-based results is missing. The benefits on the sentiment analysis task are instead considerably small.

Multiple dropout ratios have been tested, thus making the evaluation section more robust.

A theoretical analysis of the convergence of Hetero AsyncDrop is provided in the appendix.

Minor issues:

-) An additional page is inserted between the first bibliography section and the appendix.

-) One typo in the appendix (experimebts)

-) There are two bibliography sections

---

### Meta-Review · Area_Chair_ZRXv · 2023-05-05

**Recommendation:** Accept
**Confidence:** 4

**Metareview:**

Dear authors,

The reviews for the paper are included below. Overall, the reviewers positively agree on the quality of the submission, and I concur. I recommend revising the paper to address the valuable comments from the reviewers, which will definitely help improve the quality and readability of the paper.

Sincerely,